# Effect of Vascular Photobiomodulation in the Postoperative Period of Alveolar Bone Grafting

**DOI:** 10.3390/dj13050190

**Published:** 2025-04-26

**Authors:** Nicole Rosa de Freitas, Luisa Belluco Guerrini, Denise Sabbagh Haddad, Roberta Martinelli de Carvalho, Renato Yassutaka Faria Yaedú, Ana Lúcia Pompéia Fraga de Almeida

**Affiliations:** 1Postgraduate Program, Hospital for Rehabilitation of Craniofacial Anomalies, University of São Paulo, São Paulo 17012-901, Brazil; nicolefreitas@usp.br; 2Independent Researcher, São Paulo 13416-345, Brazil; lu_guerrini_@hotmail.com; 3APCD Sector of Dentistry, São Paulo 02011-000, Brazil; deniseshaddad@hotmail.com; 4Sector of Oral and Maxillofacial Surgery, Hospital for Rehabilitation of Craniofacial Anomalies, University of Sao Paulo, São Paulo 17012-901, Brazil; romartinelli@usp.br (R.M.d.C.); yaedu@usp.br (R.Y.F.Y.); 5Department of Surgery, Stomatology, Pathology and Radiology, Bauru School of Dentistry, University of São Paulo, São Paulo 17012-901, Brazil; 6Department of Prosthodontics and Periodontics, Bauru School of Dentistry, University of São Paulo, São Paulo 17012-901, Brazil

**Keywords:** cleft palate, lasers, bone transplant, pain, thermography

## Abstract

**Background/Objectives:** This study evaluated the effects of vascular photobiomodulation (VPBM) on pain intensity, edema, and facial temperature variation in patients undergoing alveolar bone grafting (ABG) surgery. **Methods:** A total of 42 patients with cleft lip and palate (aged 9–25 years) scheduled for ABG using iliac crest bone were randomly assigned in equal numbers (14 per group) to one of three groups: control (ABG only), test (ABG + VPBM), and placebo (ABG + simulated VPBM). Iliac and facial pain and edema were clinically evaluated 24 h post-surgery, along with thermographic facial analysis. Follow-up was conducted via phone calls for one week. **Results:** No statistically significant differences were observed among the groups regarding facial pain and edema at 24 h post-surgery. However, iliac pain significantly differed between the placebo and control groups (*p* = 0.045). A significant time-related effect on both facial and iliac pain outcomes was noted during follow-up, irrespective of the group. The need for rescue medication and self-perception of reduced edema did not differ significantly. Thermographic analysis reveals a significantly lower temperature variation in the test group (2.36 °C) compared to the other groups (*p* = 0.007). **Conclusions:** Overall, VPBM therapy influenced postoperative pain in the early recovery phase and temperature in the immediate postoperative period but did not significantly affect edema.

## 1. Introduction

Cleft lip and palate are congenital anomalies that develop between the 4th and 12th weeks of intrauterine life. These anomalies result from the failure of the frontal and maxillary processes to fuse properly, leading to complex regenerative challenges that necessitate a multidisciplinary approach for effective rehabilitation [1].

Alveolar bone grafting (ABG) is a common surgical intervention for cleft lip and palate, particularly in cases involving the lips and alveolus. This procedure is typically performed before the eruption of permanent canines (between 8 and 12 years of age) and is known as secondary alveolar bone grafting (SABG). When performed after canine eruption, it is termed late bone grafting, and when conducted in adulthood, it is referred to as tertiary alveolar bone grafting (TABG) [2]. The surgery involves harvesting bone tissue from the iliac crest, mandibular symphysis, or other potential donor sites and grafting it into the cleft. However, this procedure is associated with significant pain and morbidity as it involves two surgical sites [2].

Photobiomodulation therapy (PBM) has emerged as a promising adjunct to pharmacological treatments, mitigating the disadvantages associated with autogenic bone graft techniques, particularly painful symptoms. PBM is widely used for its ability to induce cellular changes that enhance metabolism, modulate inflammation, provide analgesia, and promote tissue repair [3].

Originally introduced in the Soviet Union in the 1980s as a low-intensity laser therapy for treating cardiovascular diseases, photobiomodulation systemic therapy involved an invasive technique of irradiating blood through catheters and optical fibers. To facilitate broader clinical applications, including its use in outpatient settings, this technique has since been adapted for non-invasive systemic blood irradiation through the skin called VPBM (vascular photobiomodulation) [4,5]. When administered systemically, PBM is transported via hemoglobin and helps regulate antioxidant defenses, thereby reducing oxidative stress. Moreover, VPBM modulates inflammation by lowering proinflammatory cytokine levels, which reduces edema, accelerates healing, and provides analgesic effects [3].

The effects of PBM on pain and edema have been extensively studied, typically utilizing subjective assessment tools. However, a novel approach involves using infrared thermography, a non-invasive imaging technique that captures infrared radiation emitted by the body. This method enables objective measurement of temperature changes associated with signs and symptoms of inflammatory processes [6]. To date, the use of infrared thermography to evaluate various cardinal signs of inflammation in patients undergoing SABG or TABG has not been explored.

This study aimed to evaluate the additional effects of VPBM on pain intensity, edema, and facial temperature variations following alveolar bone graft surgery. This study aims to support the hypothesis that VPBM reduces inflammation and mitigates its cardinal signs.

## 2. Materials and Methods

This study was submitted to and approved by the Ethics Committee on Human Research of the Hospital for Rehabilitation of Craniofacial Anomalies, University of São Paulo (CAAE: 01436618.4.0000.5441. Trial registration number: ID RBR-2trzh45, 10 October 2023, Adjunct effect of Transdermal Systemic Photobiomodulation in Alveolar Bone Graft Surgery).

A convenience sampling method was used, selecting patients based on the demand for bone graft surgeries, ensuring the representativeness of the target population. The statistical power analysis indicates a power of only 4.9%, markedly below the conventional threshold of 80%, potentially reflecting the casuistic nature of the study population.

Patients were recruited at the maxillofacial surgery clinic following a meticulous evaluation by the surgical team to confirm their suitability for the surgical procedure. All patients and their legal guardians (for those under 18 years) were informed about the study, received detailed explanations, and provided informed consent after having their questions addressed.

The inclusion criteria were patients having bilateral cleft lip and palate and being candidates for ABG using an iliac crest graft. A total of 62 patients were initially selected.

Exclusion criteria included declining participation and/or withdrawal at any stage of the study (three patients), testing positive for Severe Acute Respiratory Syndrome Coronavirus 2 (one patient), modification in the surgical plan during the procedure (eight patients), the presence of facial hair (beard or mustache) at the time of imaging, as it could compromise facial temperature analysis [7] (three patients), patients with dark skin complexion due to a higher melanin content, which absorbs infrared laser energy [6] and may increase the risk of burns (two patients), and patients with systemic conditions preventing ABG (three patients).

The final sample included 42 patients aged 9–25 years, who were randomly assigned to three groups (*n* = 14 per group). In the control group, only ABG surgery was performed. The test group received VPBM, whereas the placebo group received a simulated therapy. Data were compiled into a table and analyzed using simple linear regression (Jamovi Software, version 2.2.3, Sydney, Australia). Initially, the final outcomes (pain, edema, and temperature) were assessed for correlation with surgical performance predictors, including procedure time, age, and sex, with no statistically significant differences found (*p* > 0.05). Consequently, all patients were grouped based on the treatment received.

ABG was performed by two experienced maxillofacial surgeons from the Hospital for Rehabilitation of Craniofacial Anomalies, following the protocol described by Boyne and Sands (1972) and later applied by Oslo [2].

Surgeries were performed under general anesthesia, with doses adjusted according to each patient’s needs. Additionally, 1% xylocaine with a vasoconstrictor was administered to minimize intraoperative bleeding. The surgical technique for bilateral ABG involves an oblique buccal incision between the center of the first molar crown and the mesial gingival papilla on both sides. This is followed by an intrasulcular incision that extends to the lateral margin of the clefts, contouring the gingival limits and ending in the intrasulcular region of the central incisors on both sides. A full-thickness mucoperiosteal flap is then detached from this initial oblique incision, maintaining a midline buccal pedicle. Next, the nasal floor mucosa is detached, repositioned superiorly, and sutured to close the buconasal fistula, creating a physical space for graft placement. The palatal mucosa is also dissected and sutured. The bone graft, harvested from the iliac crest by a plastic surgeon and properly particulated, is carefully placed into the defect. The vestibular flap is then repositioned until it completely covers the grafted bone and entire bone extension without tension. Finally, the incision edges are debrided, and the incisions are sutured with simple stitches.

Preoperatively, patients received the following intravenous medications: cefazolin (1 g or 2 g for those weighing > 50 kg), metronidazole (500 mg), dexamethasone (2 mg/mL), and metamizole sodium (dipyrone, 2 mL). Dipyrone, an analgesic and antipyretic widely used in Brazil, is not approved in several countries.

Postoperatively, patients were prescribed intravenous cefazolin (1 g), ketoprofen (100 mg) every 8 h, dipyrone (2 mL) every 6 h for 24 h, and one drop of 0.9% saline solution in each nostril four times daily. All medications were administered according to each patient’s weight and age.

VPBM was administered immediately after surgery by a trained operator (N.R.F) approximately 10 min postoperatively while the patient was still sedated in the surgical center’s recovery room and again 24 h after the procedure in the research room. These procedures were performed using the Laser TherapyEC device (DMC Import and Export of Equipment Ltd., São Carlos, São Paulo, Brazil), which operates in the red spectrum (660 ± 10 nm), with a power output of 100 mW ± 20% in the intravenous laser irradiation of blood (ILIB) mode. The therapy duration was 10 min for patients aged 9–17 years and 15 min for those aged ≥18 years. VPBM was applied to the radial artery of the arm opposite to that used for medication administration using a specialized wristband provided by the device manufacturer. (Figure 1).

Blood pressure was monitored every 5 min in the surgical clinic using a multiparameter monitor (G3G, General Meditech Inc, Shenzhen, Guangdong, China) and in the outpatient setting using an automatic digital arm device (G-TECH Home BSP11, G-Tech, Accumed-Glicomed, Londrina, Paraná, Brazil). Oxygen saturation was measured every minute in the surgical clinic using a pulse oximeter (Oxypleth, Dixtal Biomédica Industry and Commerce Ltd., Barueri, São Paulo, Brazil) and in the outpatient setting using a wrist oximeter (OLED G-Tech, G-Tech, Accumed-Glicomed, Londrina, Paraná, Brazil). Monitoring was conducted throughout all photobiomodulation therapy sessions. For the simulated therapy application, the device was turned on to emit sound signals; however, the button activating the therapeutic light was not pressed.

Postoperative pain was evaluated using the Visual Analog Scale (VAS) 24 h post-surgery. Patients were asked to mark a vertical line on a 100 mm horizontal line corresponding to their pain level for both the facial and iliac regions.

After discharge, the pain was evaluated daily for up to 7 days post-surgery using the 4-point Verbal Evaluation Scale (VRS-4) via telephone (WhatsApp messages). Patients or their guardians reported current pain levels and whether rescue medication was required. The VRS-4 [8] used scores for pain as follows: 0 = no pain, 1 = mild pain, 2 = moderate pain, and 3 = severe pain.

Edema was evaluated using a metric tape with reference points adapted from Piso et al. [9]. The following distances were recorded in millimeters: (1) tragus to the wing of the nose, (2) tragus to the lip corner, (3) maxillary angle to the inner corner of the eye, and (4) maxillary angle to the cleft scar (bilaterally). Measurements were recorded preoperatively (T0) to establish a baseline for each patient and in the immediate post-surgical stage (T1) to assess edema severity.

Measurements were performed by a trained operator, starting with 20% of the total sample and achieving an interclass correlation coefficient (ICC) of 0.874 (IBM^®^ SPSS^®^ Statistics Software, version 22, Armonk, New York, NY, USA), which was considered good [10]. The total measurements from both sides were averaged to evaluate facial edema. The edema coefficient was calculated in millimeters using the following formula [11]:Coefficient of edema = (postsurgical face − presurgical face)/(presurgical face) × 100

Following discharge, patients were asked every 48 h for up to 7 days to rate their perceived reduction in facial edema. Patients quantified this perception using the following scores: 0 = no reduction, 1 = slight reduction, 2 = moderate reduction, and 3 = considerable reduction.

Thermographic images were captured using a FLIR Thermographic Camera T2-T540SC (176,800 pixels, sensitivity of 30 mK at 30 °C, 5 MP visual camera, 24° lens, 10 mm; Teledyne FLIR LLC, Wilsonville, OR, USA). Images were acquired preoperatively (T0) and 24 h postoperatively (T2). To ensure consistency and minimize variables affecting image acquisition, a standardized protocol was followed in accordance with the Infrared Thermography for Systemic Oral Health guidelines [7]. This protocol accounts for environmental, individual, and technical factors.

The camera was positioned perpendicularly, 80 cm from the patient’s face, using a tripod. To standardize facial positioning during image capture, a guideboard was made using acetate (8.3 cm × 6.3 cm), with vertical and horizontal lines dividing it into nine rectangles (2.7 cm × 2.1 cm). For frontal images, alignment was based on reference lines corresponding to the field of view and chin region (Figure 2a). For lateral images, guidelines were aligned with the outer corner of the eye to ensure consistency (Figure 2b). Each patient was photographed from frontal, right-side, and left-side perspectives. During the evaluation, patients were instructed to remain still, avoid touching their faces, and refrain from speaking for 15 min to achieve thermal equilibrium.

Images were processed using VisionFy software (Thermofy Consultoria em Informática Ltd., São Paulo, Brazil). Regions of interest (ROIs) were selected based on the guidelines of a previous study by Haddad et al. [12], with additional ROIs incorporating cleft scars in the frontal view. The ROIs used for the right and left sides (Figure 3a,d) were as follows: R1 and R2 (eyelid lift), R3 and R4 (nasolabial folds), R5 and R6 (lip balm, suitable for fissure scars), R7 and R8 (lip cleft), and R9 and R10 (lower lip). For lateral images, a trapezoidal ROI corresponding to the surgical site was used on both sides (Figure 3b,c,e,f).

Temperature measurements were performed by a single operator specifically trained for the study, achieving an ICC of 0.868. Each ROI provided an average temperature. The total facial temperature was calculated as the average of all frontal ROIs combined with lateral ROIs at baseline (T0) and 24 h postoperatively (T2).

The data were tabulated and analyzed in the program Jamovi Statistician (version 2.2.3) and IBM^®^ SPSS^®^ Statistics software, version 22 (Armonk, New York, NY, USA).

Sex distribution across groups was evaluated using the chi-square test (*p* < 0.05). Age and duration of the surgical procedures were analyzed using the analysis of variance (ANOVA) test (*p* < 0.05).

Facial pain (24 h post-surgery) was analyzed using the Kruskal–Wallis test (*p* < 0.05). Iliac VAS scores were assessed using ANOVA, followed by Tukey’s test for intergroup comparisons. Edema variables were analyzed using ANOVA variances (*p* < 0.05). For weekly patient follow-ups, pain and edema perceptions were analyzed. Patient-reported scores were averaged, and a two-way repeated-measures ANOVA was conducted [13].

Verbal pain assessments for the face and iliac regions, as well as perceived edema reduction, were analyzed using repeated-measures ANOVA, followed by Fisher’s Least Significant Difference (LSD) test (ɑ = 0.05). The use of rescue medication, measured by the number of days each patient required pain medication, did not follow a normal distribution and was therefore analyzed using the Kruskal–Wallis test (*p* < 0.05).

Facial temperature was evaluated by calculating the temperature difference (delta value) between T2 and T0 across groups using ANOVA, followed by Tukey’s test (*p* < 0.05). The correlation between facial pain, edema, and temperature was analyzed using Spearman’s correlation coefficient (*p* < 0.05), while the correlation between edema and facial temperature was analyzed using Pearson’s correlation coefficient (*p* < 0.05).

## 3. Results

The average surgery duration was 138, 146, and 127 min for the control, test, and placebo groups, respectively, with a mean duration of 137 min. No statistically significant differences were observed in age and procedure time among the different study groups (*p* = 0.785 and *p* = 0.146, respectively).

The median VAS score for facial pain was lower in the test group (5.0) than in the placebo (11.5) and control groups (7.0); however, this difference was not statistically significant (*p* = 0.339). In contrast, the VAS score for iliac pain showed a statistically significant difference (*p* = 0.045) only between the placebo (52.28 ± 22.91) and control (28.71 ± 23.05) groups. Edema, measured 24 h post-surgery, did not differ significantly among groups (*p* = 0.634).

During the first postoperative week, time was the only factor influencing facial pain, iliac pain, and edema perception.

For facial pain (Figure 4), the most notable difference occurred at 96 h, where the test group (0.14) showed a statistically significant reduction compared to the control group (0.78).

Iliac pain was significantly reduced at 72, 96, and 120 h, regardless of treatment (Figure 5). After this period, no statistically significant difference was observed (0.5 at 144 h and 0.38 at 1 week).

For edema (Figure 6), a longer evaluation period correlated with a greater perception of edema reduction, regardless of treatment.

The number of days patients required rescue medication was analyzed using the median (control = 1, test = 1, and placebo = 2), with no statistically significant difference (*p* = 0.607).

Facial temperature variations among groups were evaluated using the delta value (T2-T0). The test group exhibited a significantly lower temperature variation (2.36 °C ± 0.94; *p* < 0.05) compared to the control (3.35 °C ± 0.81) and placebo (3.30 °C ± 0.88), with no statistically significant difference between the latter two.

Correlation analysis between pain, edema, and facial temperature regardless of treatment (T2) shows no significant associations (edema–temperature: *p* = 0.173, r = 0.214; VAS of the face–edema: *p* = 0.218, r = 0.194; VAS of the face–edema–temperature: *p* = 0.298, r = −0.164).

## 4. Discussion

This study evaluated the effects of VPBM on pain, edema, and facial temperature. In the short term (24 h post-surgery), the therapy was not effective. However, weekly analyses reveal a reduction in facial pain in the test group, and thermographic analysis shows significant results supporting the therapy’s efficacy.

The industry-recommended VPBM dose duration is 30 min, although no studies in the literature strongly support this protocol. In this study, the therapy duration was reduced to 10 min for children (one-third of the recommended time) and 15 min for adults (half of the recommended time), based on a study by Moskvin et al. [14]. As the first applications were administered with the patients still under general anesthesia, the dose was intentionally reduced due to the unknown interactions between the systemic therapy and the prescribed anesthetic medications. Throughout the procedure, patients’ vital signs were monitored, and no VPBM-related adverse events, such as increased blood pressure, oxygen desaturation, or post-surgical bleeding, were observed.

For the iliac pain, evaluated within 24 h post-surgery, VPBM did not yield significant results. A significant difference was observed between the control (28.71 ± 23.05) and placebo (52.28 ± 22.91) groups. This result may be attributed to the sample size or the possibility that more pain-sensitive patients were randomly assigned to the placebo group. Since patients were sedated during this phase, the potential psychological effects of the simulated therapy could not be considered.

Similarly, VPBM did not yield significant results for facial pain within 24 h (test vs. control and test vs. placebo), likely due to insufficient time for biomodulation during the evaluation period. This corroborates the findings of D’Avilla et al. [15], who, despite evaluating the effects of the local PBM on patients undergoing orthognathic surgery, found no significant results for pain in the immediate postoperative period. While VPBM and local PBM are distinct therapies with different protocols and applications, both are considered photobiomodulation therapies and are performed using the same laser modality. Therefore, it is reasonable to draw parallels between them, particularly given the lack of studies in the literature on PBMST in patients with bilateral lip and palatal clefts undergoing ABG surgery.

Ideally, to evaluate the efficacy of PBM in managing pain and edema, patients who have undergone surgery should receive only the therapy without additional medications to avoid confounding effects. However, in this study, antibiotics, anti-inflammatory drugs (both steroids and non-steroids), and analgesics were administered, with dosages calculated based on individual body weight. Another factor that may have influenced pain evaluation was the timing of medication administration. Some patients completed the VAS assessment while under the effect of analgesics, but adjustments to this variable were not possible due to hospital protocol. The use of medication is a limitation in human studies as it is administered before or after procedures (e.g., analgesics, antibiotics, and anti-inflammatory drugs), potentially obscuring the true effects of PBM [16,17].

Facial pain assessment using VAS scores reveals a statistically significant difference between the test (0.14) and control (0.78) groups at 96 h of evaluation. By 120 h, no pain was observed in the test group. The mechanism underlying the positive results of VPBM involves light absorption by hemoglobin, which activates the superoxide dismutase enzyme. This enzyme interferes with the arachidonic acid cascade, inhibiting prostaglandin production and, consequently, the release of chemical mediators such as serotonin and bradykinin, which activate nociceptors and induce pain [18].

At 48 and 72 h, significant differences were observed between the control (0.85 and 0.78, respectively) and placebo groups (0.42 and 0.35, respectively). These results suggest that the group receiving simulated therapy exhibited the lowest average pain scores at both time points. This finding indicates a potential placebo effect, which may be attributed to psychological factors based on the patient’s belief in the effectiveness of VPBM (17) or due to the possibility that placebo therapy stimulated the release of endogenous opioids, which themselves influence pain perception [19].

Although VPBM effectively reduced facial pain, its impact could have been enhanced if the therapy sessions had been performed throughout the week, optimizing the scores for iliac pain, which was not influenced by the therapy, as the pain diminished throughout the week regardless of the treatment. The cumulative effect of photobiomodulation is well established; however, there is no consensus in the literature regarding the optimal number of sessions required for clinical efficacy [20]. In this study, hospital regulations limited PBM applications to 10 min for children and 15 min for adults, which may have been insufficient to achieve maximum therapeutic benefit.

Pain control during the week was also evaluated based on the number of days each group required rescue medication. No statistically significant differences were observed among the groups, indicating that regardless of the treatment employed, patients who were more sensitive to pain received medication for more days. These results differ from those reported by Ezzat et al. [21], who reported reduced reliance on rescue medication on the second and third days in the PBM-treated group. However, their methodology involved localized PBM application, which may explain the discrepancy.

Photobiomodulation can reduce edema by directly or indirectly influencing the lymphatic system and enhancing its function [22]. However, in this study, the edema coefficient did not differ significantly among the groups. Edema typically peaks between 48 [11] and 72 [23] h post-surgery; however, due to study limitations, our analysis was performed at 24 h intervals. In the self-perception verbal scale assessing edema reduction, only time showed a significant interaction with the results. Many patients reported no swelling at the end of the study, a fact directly related to the evolution of edema, which completely resolved between 5 and 7 days [11].

Pain assessment is inherently subjective, making it challenging to quantify. Thermography, however, is a potent tool for objectively evaluating postoperative inflammation [24]. In this study, the test group exhibited a lower temperature variation in the delta value, suggesting an anti-inflammatory effect of PBM. Our findings partially corroborate those of Pedreira et al. [16], who, during the evaluation of the effect of PBM after third molar exodontia surgery, also noticed a reduction in temperature in certain ROIs. As their study employed a different methodology, laser type, and a distinct selection of ROIs, comparisons should be made cautiously. Nonetheless, it cannot be excluded that PBM may have positively influenced the inflammatory process following surgical trauma.

Analysis of correlations between pain, edema, and facial temperature reveals a weak or nonexistent relationship, corroborating the findings of Christensen et al. [24], and yielded no statistically significant results. According to the 19th edition of the *Pan American Journal of Medical Thermography* [7], steroids and other medications should ideally be avoided 12–16 h before examination. However, since this study was conducted in a hospital setting, it was not possible to modify patients’ medication regimens before the examination. Consequently, all patients received steroids and analgesics before thermographic imaging post-surgery. Another challenge was the inconsistency in the timing of thermographic imaging across groups.

Given the limited literature on VPBM, establishing a standardized safety protocol remains a significant challenge. Additional randomized clinical trials are required to refine protocols that ensure both therapeutic efficacy and patient safety. One of the primary limitations of this study relates to the adopted experimental protocol, particularly the restricted number of sessions implemented. Notably, the observed reduction in hospital stays highlights the importance of identifying an optimal protocol, or at a minimum, one capable of producing positive outcomes during hospitalization with fewer sessions.

Furthermore, the limited sample size may have influenced the findings. To our knowledge, this is the first reported study to investigate VPBM sessions in patients with bilateral cleft lip and palate using infrared thermography. The results support the hypothesis that VPBM modulates inflammation and its cardinal signs, primarily affecting heat and pain in the immediate postoperative period.

Nevertheless, further studies involving larger sample sizes, standardized protocols for both pediatric and adult populations, extended follow-up durations, and individualized pain assessments—aimed at modulating pre-treatment pain—are essential for generating more robust evidence regarding the therapy’s efficacy and ensuring its accurate and safe clinical implementation.

## 5. Conclusions

Overall, VPBM therapy influenced postoperative pain in the early recovery phase and temperature in the immediate postoperative period but did not significantly affect edema.

## Figures and Tables

**Figure 1 dentistry-13-00190-f001:**
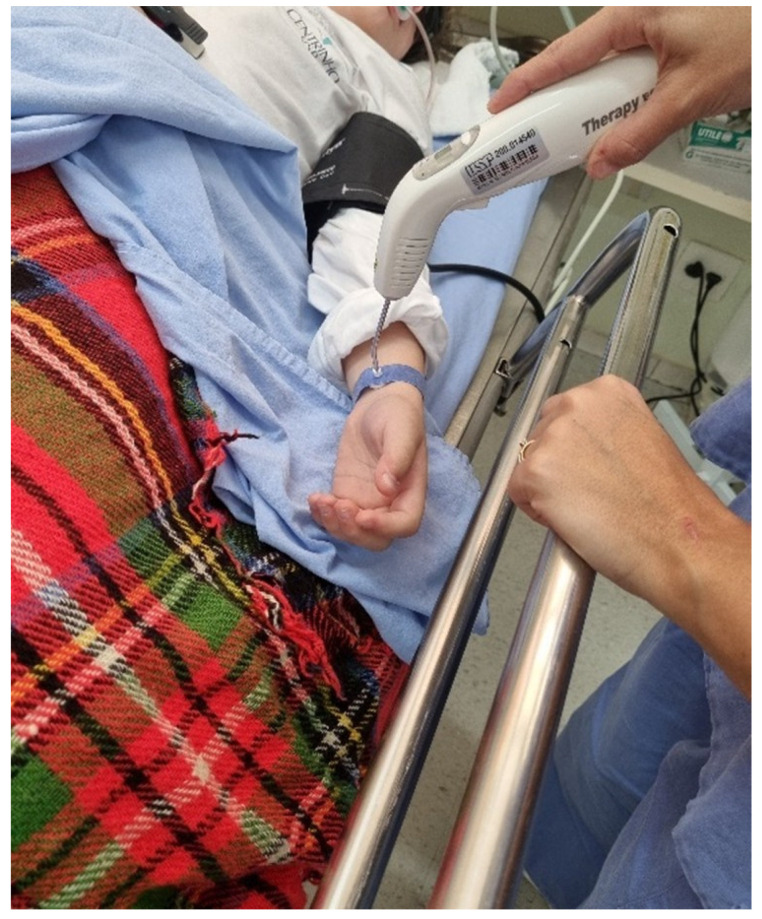
Application of vascular photobiomodulation to the radial artery of the arm opposite to that used for medication administration using a specialized wristband provided by the device manufacturer.

**Figure 2 dentistry-13-00190-f002:**
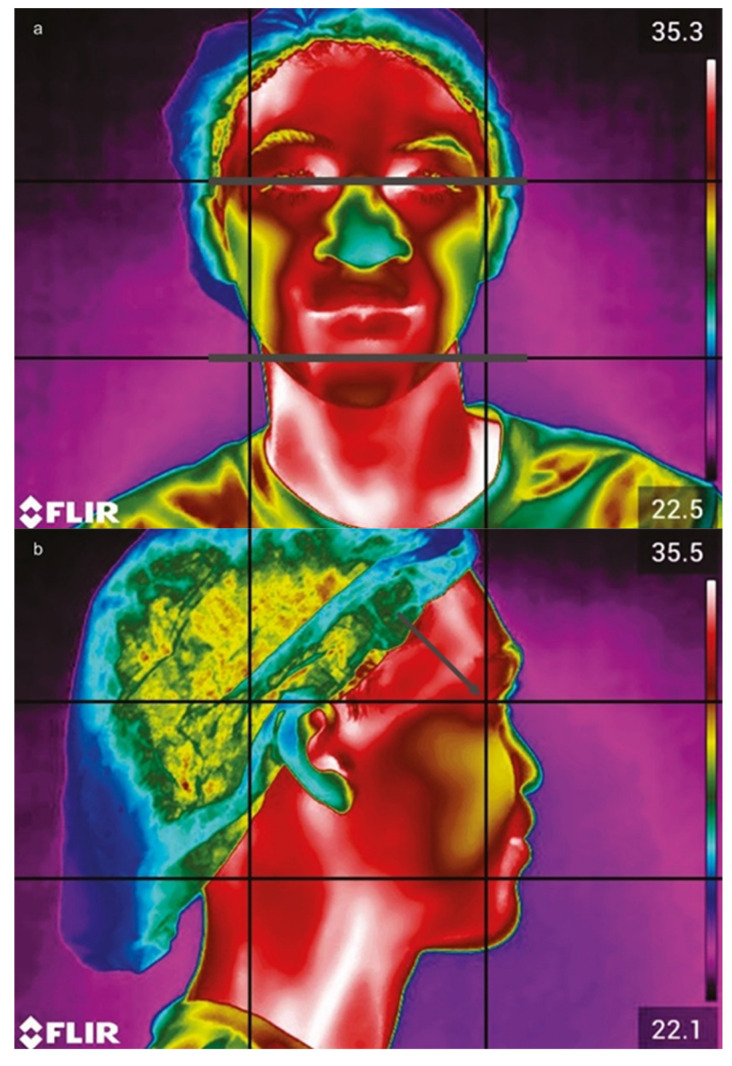
Camera view focusing on the frontal plane, with gray lines indicating the bipupilled view (**a**). In the side position (**b**), the guidelines are aligned with the region of the corner of the eye (gray arrow).

**Figure 3 dentistry-13-00190-f003:**
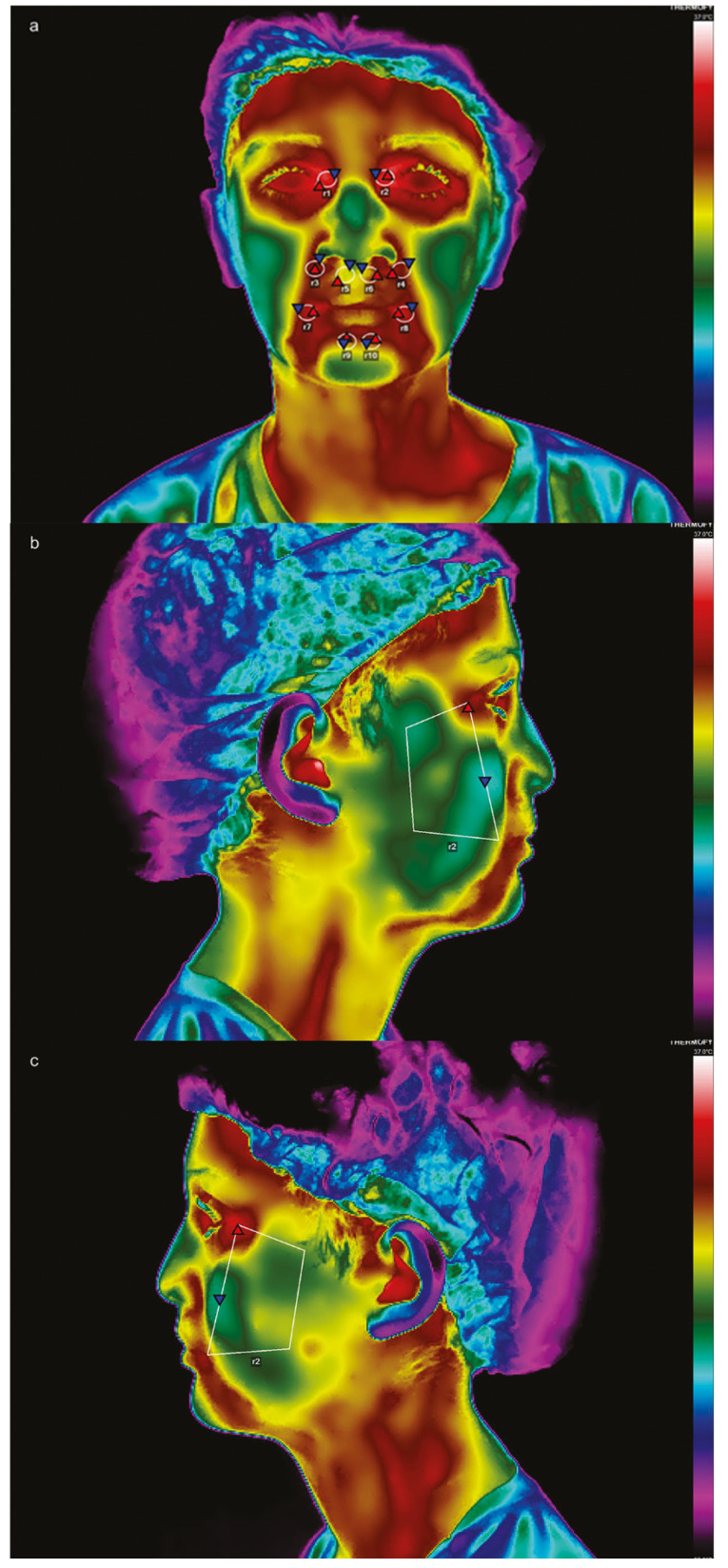
Frontal view in thermographic imaging before (**a**) and after (**d**) surgery; side views before (**b**,**c**) and after surgery (**e**,**f**), with their respective regions of interest (ROIs).

**Figure 4 dentistry-13-00190-f004:**
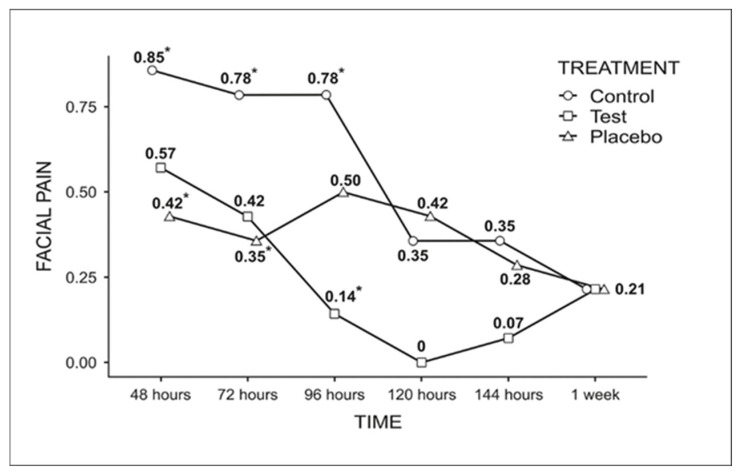
Graph of the comparison among the groups for facial pain analysis throughout the week (Fisher’s LSD test, *p* < 0.05). * Indicates statistically significant difference.

**Figure 5 dentistry-13-00190-f005:**
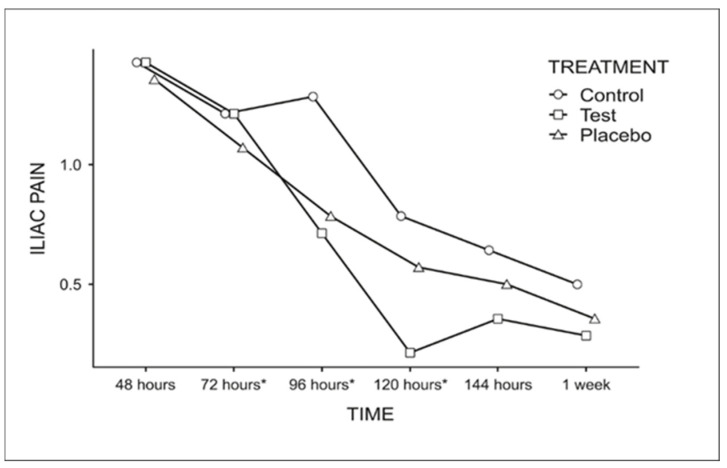
Graph for the comparison of the average score for iliac pain among the evaluation times, regardless of the treatment used (Fisher’s LSD test *p* < 0.05). * Indicates statistically significant difference.

**Figure 6 dentistry-13-00190-f006:**
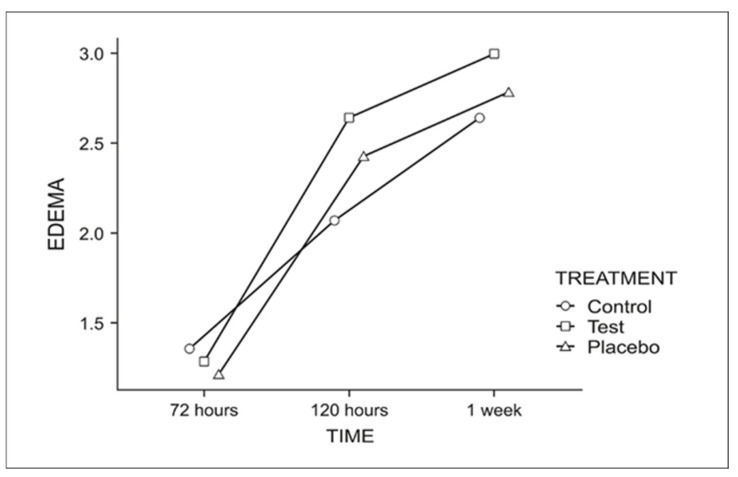
Graph for the perceived edema reduction throughout the week.

## Data Availability

The raw data supporting the conclusions of this article will be made available by the authors on request.

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
