# Peer review of "Effect of Vascular Photobiomodulation in the Postoperative Period of Alveolar Bone Grafting"

_dentistry, 2025, doi:10.3390/dj13050190_

Round 1
Reviewer 1 Report
Comments and Suggestions for Authors
- General concept comments
Article: The manuscript is very clear to understand as per the following
- The introduction is well structured by addressing each aspect of the study, trying to correlate each one of them (SABG, TABG, PBM,VPBM and Thermography) deriving a hypothesis and an aim for the study, thereby creating relevance.
- The materials and methods proceeded with approval from the ethics committee.
The sample exclusion criteria is very detailed to avoid bias in the study. The methodology has been previously employed and cited. The figures, table and images are appropriate.
- Selecting 3 criteria of the cardinal signs of inflammation (Pain intensity, edema and facial temperature variations) was a good approach as multiple variables provide more insight into the effectiveness of VPBM.
- The discussion in particular is very well written and sound. The authors discussed each factor (medications, timing of administration, hospitalization, sedation) comparing their study with previous ones identifying the similarities and differences in the results owing to the difference in methodological specifications.
- The cited references are all relevant and support the manuscript.
- The conclusions were significant and aligned with the aim of the study.
- Specific comments
- Line 54-
It was Dr. Endre Mester in 1967 at the Semmelweis Medical University (Hungary) who first described the effects of low level laser irradiation in mice. Mester applied light from a ruby laser (694 nm) in an attempt to cure malignant tumors both in rats and also tested it in human patients. Unfortunately, Mester failed to cure any tumors, but did observe a faster rate of hair growth in the treated mice compared to the controls, calling this effect "laser biostimulation".
McGuff PE, Deterling RA, Gottlieb LS. Tumoricidal effect of laser energy on experimental and human malignant tumors. N. Engl. J. Med. 1965;273(9):490–492. doi: 10.1056/NEJM196508262730906.
Mester E, Szende B, Gärtner P. The effect of laser beams on the growth of hair in mice. Radiobiol. Radiother. (Berl) 1968;9(5):621–626.
Kovács IB, Mester E, Görög P. Stimulation of wound healing with laser beam in the rat. Experientia. 1974;30(11):1275–1276
- It would be beneficial to include the parameters on which the effectiveness of PBM on the target tissue such as light source, wavelength, energy density, light pulse structure, and the duration of the laser application.
Kujawa J., Pasternak K., Zavodnik I., Irzmański R., Wróbel D., Bryszewska M. The effect of near-infrared MLS laser radiation on cell membrane structure and radical generation. Lasers Med. Sci. 2014;29:1663–1668.
- Using Metamizole an analgesic and antipyretic may have been a disadvantage as it is not used in some countries due to the risk factors associated as these results may not be applicable to some populations. Choosing an analgesic that is used worldwide would have enhanced the reach.
- Line 132- Laser Classification and Type could have been mentioned. The Laser Therapy EC Device used is a Semiconductor Laser InGaAIP.
- Pertaining to Line 305 – Conti et.al in 1997, Masse et.al in 1993 and Ribeiro et.al in 2008 evaluated the effect of LLLT in patients without any other medications as confounding factors. This is probably possible owing to the non-invasive nature of their studies where they mainly assessed Temporomandibular Disorders and Periodontal therapy.
Conti P.C. Low level laser therapy in the treatment of temporomandibular disorders (TMD): A double-blind pilot study. Cranio. 1997; 15: 144-149.
Ribeiro I.W.J., Sbrana M.C., Esper L.A., Almeida ALPF. Evaluation of the effect of the GaAIAs laser on subgingival scaling and root planing. Photomed Laser Surg. 2008; 26: 387-391.
Masse J.F., Landry R.G., Rochette C., Dufour L., Morency R., D'Aoust P. Effectiveness of soft laser treatment in periodontal surgery. Int Dent J. 1993; 43: 121-127.
Author Response
- Authors have to describe in details the Photobiomodulation treatments in the Section: "Material and methods". What were the power, dose, location, number of treated points, etc. Photos have to be added for one treated case showing the different treated points.
To treat systemic photobiomodulation, lasers are not applied to the face but rather to specific blood vessels in the human body. Blood irradiation can be performed using either the intravascular method (an invasive technique) or a transcutaneous application over an artery (a non-invasive technique, referred to in this study as vascular photobiomodulation). In this study, the irradiated region was the radial artery. However, other studies have explored applications in the popliteal, carotid, and sublingual regions, among others.
Information regarding application, power, and wavelength is provided between lines 134 and 142.
An image of the therapy being performed is attached to the manuscript.
- The Photobiomodulation was only done by red laser light or infrared (deeper penetration)? or both?
Vascular photobiomodulation is a low-level laser modality in the red spectrum of light. Only irradiation in the red spectrum was performed.
- The PBM was done exclusively in oral cavity or also on the area of the iliac crest bone? There is a loss of detailed description in the manuscript for both treated areas by PBM.
Vascular photobiomodulation was not performed in the oral cavity or iliac crest. As it is a systemic application, the application is carried out in large vessels and the benefits of photobiomodulation are transported via hemoglobin to different areas of the human body. In the present study, the application was carried out in the radial artery and only clinical analyzes of pain and edema in the oral cavity, as well as pain in the iliac crest region, were carried out. The present study aimed to evaluate whether the application of this laser modality, with this route of application (in the region of the radial artery) would be useful to reach these two areas involved in the alveolar bone graft procedure, which has two access areas (the intra-oral region, in the fissure region, as the graft receiving area) and the iliac cyst region as the graft donor area.
Vascular photobiomodulation was not performed in the oral cavity or iliac crest. Since this is a systemic application, it is administered in large blood vessels, allowing the benefits of photobiomodulation to be transported via hemoglobin to different areas of the human body. In the present study, the application was performed on the radial artery, and only clinical analyses of pain and edema in the oral cavity, as well as pain in the iliac crest region, were conducted.
This study aimed to evaluate whether the application of this laser modality via this specific route (in the radial artery region) would be effective in reaching the two areas involved in the alveolar bone graft procedure. This procedure includes two access sites: the intraoral region (in the fissure region, as the graft recipient area) and the iliac crest region (as the graft donor area).
- Authors have to add some photos describing the points of PBM treatments.
The photo of the treatment was attached to the manuscript.
- Trial registration number was added to the manuscript: ID RBR-2trzh45, October 10th, 2023 (Adjunct effect of Transdermal Systemic Photobiomodulation in Alveolar Bone Graft Surgery)

Reviewer 2 Report
Comments and Suggestions for Authors
The authors present a highly interesting work on the benefits that vascular photobiomodulation can provide in the recovery of patients undergoing alveolar bone graft surgical procedures. Although some of their results do not show significant differences, from my point of view, they present an elegant and well-articulated methodological approach. I only note a few minor corrections that I identify in their work:
- Standardization of signaling suffixes in figures: lowercase should be used in the images and uppercase in the descriptions.
- The use of ANOVA, Kruskal-Wallis, and correlation tests is mentioned; however, it would be helpful to clarify whether normality tests were conducted before selecting these analyses (Lines 216-233).
- The phrase "ANOVA variance" is redundant; it should simply be "ANOVA" (Line 221).
- Some comparisons describe trends despite lacking statistical significance (e.g., "the median VAS score for facial pain was lower... however, this difference was not statistically significant" in Lines 239-240). If a result is not statistically significant, avoid presenting it as a trend unless there is a strong justification.
- It is unclear whether the reported value refers to a p-value (Lines 247-249).
- Consider restructuring the results section with clearer subheadings for pain, edema, and temperature to improve readability.
- The discussion on the placebo effect could be strengthened by incorporating additional scientific references (Lines 290-294).
- The conclusion is well-articulated but could briefly suggest future research directions beyond increasing sample size.
Author Response

(The authors gave the same response as above.)

Reviewer 3 Report
Comments and Suggestions for Authors
I reviewed this interesting work on the effects of vascular photobiomodulation. The topic is relevant and using VPBM combined with thermographic analysis is innovative. However, while the methodology is generally sound, several aspects require further clarification and improvement to strengthen the validity and clinical applicability of the study.
1) The manuscript does not specify the randomization method. Clarifications are needed.
2) Regarding the VPBM protocol, the rationale behind the frequency and duration of sessions should be stated and supported by references.
3)The manuscript mentions previous studies on PBM but does not differentiate the specific benefits of VPBM compared to conventional PBM.
4) It would be appropriate to justify the sample size with an appropriate power analysis and clarify the randomization and blinding procedure.
5) The work only evaluates the immediate postoperative period (up to 7 days). Would a longer follow-up provide more information on the sustained effects of VPBM?
6) Some references do not have a DOI and have inconsistent formatting.
7) From this initial analysis it is clear that a major revision is needed
The study presents valuable insights into VPBM, but methodological clarification, better statistical interpretation and more critical discussion are needed before acceptance.
Author Response

(The authors gave the same response as above.)

Round 2
Reviewer 3 Report
Comments and Suggestions for Authors
The article is well structured and clear, but has some gaps in terms of originality and critical analysis of the results. It is recommended to revise the "Discussion" section by adding an analysis of the limitations of the study and future perspectives, as well as an improvement of the innovative element of the article.
It is suggested to add more details on the statistical analysis used and to discuss the possible statistical power.
Author Response
Dear reviewer
We would like to thank you for your comments; they were extremely valuable for improving the paper. The requested information has been included in the manuscript and is highlighted in yellow.
Reviewer 1
The article is well-structured and clear, but it presents some gaps in terms of originality and critical analysis of the results. It is recommended to revise the Discussion section, adding an analysis of the study's limitations and future perspectives, as well as enhancing the innovative aspect of the study.
It is also suggested to provide more details about the statistical analysis used and to discuss the possible power of the statistical test.
Response
- The statistical power analysis indicated a power of only 4.9%, markedly below the conventional threshold of 80%, potentially reflecting the casuistic nature of the study population. All issues related to statistical analysis have been reviewed.
- Given the limited literature on VPBM, establishing a standardized safety protocol remains a significant challenge. Additional randomized clinical trials are required to refine protocols that ensure both therapeutic efficacy and patient safety. One of the primary limitations of this study relates to the adopted experimental protocol, particularly the restricted number of sessions implemented. Notably, the observed reduction in hospital stays highlights the importance of identifying an optimal protocol—or at minimum, one capable of producing positive outcomes during hospitalization with fewer sessions.
Furthermore, the limited sample size may have influenced the findings. To our knowledge, this is the first reported study to investigate VPBM sessions in patients with bilateral cleft lip and palate using infrared thermography. The results support the hypothesis that VPBM modulates inflammation and its cardinal signs, primarily affecting heat and pain in the immediate postoperative period.
Nevertheless, further studies involving larger sample sizes, standardized protocols for both pediatric and adult populations, extended follow-up durations, and individualized pain assessments—aimed at modulating pre-treatment pain—are essential for generating more robust evidence regarding the therapy’s efficacy and ensuring its accurate and safe clinical implementation.
Best regards,
The authors
